# The Psychometric Properties of the Meaning of Home and Housing-Related Control Beliefs Scales among 67–70 Year-Olds in Sweden

**DOI:** 10.3390/ijerph18084273

**Published:** 2021-04-17

**Authors:** Yadanuch Boonyaratana, Eva Ekvall Hansson, Marianne Granbom, Steven M. Schmidt

**Affiliations:** Department of Health Sciences, Lund University, Box 157, 22100 Lund, Sweden; eva.ekvall_hansson@med.lu.se (E.E.H.); marianne.granbom@med.lu.se (M.G.); steven.schmidt@med.lu.se (S.M.S.)

**Keywords:** psychometric, perceived housing, aging, meaning of home, housing-related control beliefs

## Abstract

Background: The housing environment is important for health and well-being among older people, and it is important to consider both physical and perceived aspects of housing. Psychometrically sound scales are necessary to assess perceived housing. This study evaluated the psychometric properties of two instruments that measure perceived aspects of housing among a younger cohort of older adults in Sweden. Methods: A random sample of 371 participants aged 67 to 70 years (mean 67.9 (SD = 0.98)) was used. Participants lived in ordinary housing in the south of Sweden. Data on perceived aspects of housing were collected with the Meaning of Home Questionnaire (MOH) and the Housing-Related Control Beliefs Questionnaire (HCQ). Internal consistency, corrected item–total correlations, floor and ceiling effects, and construct validity were analyzed. Results: Cronbach’s alphas for all four subscales of MOH and two of three subscales of HCQ had acceptable levels (α > 0.50). Some items from both scales had low item–total correlations. All subscales, except for one from HCQ, had good construct validity. Conclusion: While both instruments had some limitations, all subscales with one exception had adequate psychometric properties. When used in different national contexts, further development may be necessary to achieve conceptual equivalence.

## 1. Introduction

The housing environment plays an important role in health and well-being, particularly among the aging population [1], and it is important to not only consider the design of the physical environment but also how people perceive their home [2,3]. While much research has been conducted related to the physical design, less focus has been placed on the perception of the environment. Some high-quality qualitative studies have been conducted capturing participants’ experiences via in-depth interviews and observations [4,5,6]. While these studies give nuanced understanding, they are usually small and do not allow generalizability on a population level. Thus, part of the challenge is the lack of adequate scales to quantify perceived aspects of housing in a larger context. Perceived housing is an overarching term used to capture different aspects of a person’s experiences of interacting and identifying with his or her home environment. Perceived housing among older people is covered by a set of theoretically informed concepts, which were developed into a four-domain model including housing satisfaction, usability in the home, meaning of home and housing-related control beliefs [7]. Instruments were developed to assess each of the domains, and these instruments were translated into six languages, including Swedish, and used in the European ENABLE-AGE project with older adults aged 80–89 years [8]. Perceived housing in later life includes behavioral, emotional and cognitive adaptations a person makes in order to successfully interact with the home environment. These perceived aspects of person–environment dynamics are crucial for understanding how the home environment is related to well-being [9,10]. In this paper, we focus on the instruments that measure meaning of home and housing-related control beliefs.

Meaning of home is one important aspect of perceived housing that focuses on emotional attachment to the home. The concept covers meaningful habits, social contracts, evaluations, goals, values, cognition and emotions of people that they relate to their home [9,11]. One way of measuring attachment to the home is the Meaning of Home Questionnaire (MOH). The MOH includes behavioral, cognitive, emotional and social aspects that represent the meaning evolved by experiences of being and living in a particular home [7]. Therefore, MOH covers four subscales, including physical, behavioral, cognitive/emotional and social scales with a broad set of questions within each scale to cover the complexity of each domain [7].

Housing-related control beliefs also represent an important aspect of perceived housing that consists of the individual’s perception of different forms of control related to the home [7]. External control is strong when an individual believes that events at home are due to luck, chance, fate or powerful others (i.e., other people who have some level of control over aspects of housing), and internal control is strong when individuals believes that events at home are controlled by their own actions [12]. This construct has been measured in previous studies using the Housing-Related Control Beliefs Questionnaire (HCQ), which measures the psychological dimensions of internal control and two aspects of external control: powerful others and chance [12]. Housing-related control beliefs have been found to be related to independence in activities of daily living at home [13], as well as making decisions about staying in place or relocating to special housing [14].

A previous study of very old people in three European countries indicated acceptable reliability and validity of the MOH and the HCQ when included in a perceived housing model that covered housing satisfaction, usability in the home, meaning of home and external housing-related control beliefs [7]. Recently, a study on psychometric properties of external housing-related control beliefs among people with Parkinson’s disease showed that reliability was acceptable in terms of the internal consistency, construct validity and corrected item–total correlation in this group. The validity was found to be good in terms of convergent and known group validity. No floor or ceiling effects were found in the questionnaire with this population [15]. In addition, the authors suggested that for a frail population that is dependent on help managing activities of daily life at home, such as older adults with Parkinson´s disease, a 14-item version of the External Control subscale would improve reliability and validity [15]. Hence, the experience of perceived housing seems to vary across different populations, supporting the need to conduct psychometric evaluation with other groups.

With new cohorts growing older, it is likely that lifestyle differences between generations also influence older adults’ perceptions of the home. To capture differences between younger and older adults’ perceptions, testing and improving reliability and validity of perceived housing instruments are of utmost importance. Therefore, this study will assess the psychometric properties of these instruments that measure perceived aspects of housing in a younger cohort of older adults in Sweden, which in the long run can lead to improved possibilities for aging-in-the-right place in later life.

Psychosocial constructs are often experienced differently in different cultures and in relation to cultural shifts across generations, so it cannot be assumed that a measurement scale will perform equally well in these different contexts [16]. Therefore, instruments designed to measure such constructs should be re-evaluated when used with a new population to account for cultural shifts. Periodic re-evaluation can be useful to ensure instruments are measuring constructs in a valid and reliable manner within a specific population. In this paper, we evaluate two instruments originally developed in Germany and now used in Sweden that measure different aspects of perceived housing in the aging population. After translation into Swedish, the instruments were used among older adults aged 80–89 years [7], but evaluation of these instruments among younger older adults is lacking.

The objective of the study was to determine if two instruments about perceived aspects of housing measure the intended constructs in a consistent manner. We therefore evaluated the psychometric properties of the Meaning of Home Questionnaire and the Housing-Related Control Beliefs Questionnaire in a younger cohort of older adults in Sweden. In particular, we tested reliability and discriminant validity among the subscales of these questionnaires.

## 2. Materials and Methods

### 2.1. Study Context and Participants

For this psychometric evaluation, data on perceived aspects of housing were utilized from the Home and Health in the Third Age Project, which aims to study the relationships between different aspects of home and health as people age. The database had information on 371 participants who lived in ordinary housing in the south of Sweden [3]. Participants were recruited from the 66-year-old cohort of a longitudinal study, the Swedish National Study on Aging and Care (SNAC)—Good Aging in Skåne (GÅS) project [17]. Data collection was made in home visits by experienced project administrators who underwent project-specific training. For more detailed information on recruitment and data collection, see Kylén et al. [3]. The Home and Health in the Third Age Project was approved by the Ethical Board in Lund (2010/431). All participants gave signed, informed consent.

Participants’ ages ranged from 67 to 70 years with a mean 67.9 (SD = 0.98) years, and 57.1% were women. The majority of participants lived with a partner (64.2%) and in multi-dwelling housing. On average, the participants had lived in the current dwelling for 19.2 years (SD = 14.2). Further information about the participants is presented in Table 1.

### 2.2. Instruments and Data

#### 2.2.1. Meaning of Home Questionnaire

MOH has 28 items divided into four subscales [3,7]. The Behavioral subscale includes six items, for example, ‘‘Being at home means for me being able to do whatever I please’’. The Physical subscale includes seven items, for example, ‘‘Being at home means for me living in a place that is comfortable and tastefully furnished’’. The Cognitive/Emotional subscale has 10 items, for example, ‘‘Being at home means for me being familiar with my immediate surrounding’’. The Social subscale includes five items, for example, “Being at home means for me meeting family, friends, and acquaintances”. Participants rated each item from 0 (strongly disagree) to 10 (strongly agree). Eight of the items were reverse scored before calculating subscale scores. As each subscale has a different number of items, scores were calculated as the mean of the items within the subscale, and higher scores indicated a stronger meaning attributed to the home [7].

#### 2.2.2. Housing-Related Control Beliefs Questionnaire

HCQ included three subscales: Internal control refers to an individual’s belief that events at home are controlled by their own actions (eight items), for instance, “Everything in my home will stay the way it is, no one is going to tell me what to do”. External control–powerful others indicates that some other person is responsible (eight items), such as “Other people have told me how to arrange the furnishing in my home”. External control–chance indicates that housing-related outcomes happen by luck, chance or fate (eight items), for instance, “Where and how I live has happened more by chance than anything else.” Participants rated each statement on a five-point scale ranging from 1 (strongly disagree) to 5 (strongly agree) [7]. A sum score was calculated for each subscale. Higher scores on Internal Control indicated more perceived personal control over the home while higher scores on External Control subscales indicated more perceived control based on chance or powerful others [3].

#### 2.2.3. Descriptive Questions

Descriptive data on sample characteristics were collected as well, such as gender, age, marital status, education, type of dwelling, number of years they had lived in the current dwelling, number of rooms in the dwelling, perceived health, and dependence on help in activities of daily living (ADL).

### 2.3. Data Analyses

Sample characteristics were described with mean and SD for continuous variables and frequencies and percentages for categorical variables. All data analyses were conducted with IBM SPSS statistic version 26 (IBM Corp., Armonk, NY, USA). 

#### 2.3.1. Reliability

We evaluated each item to determine the relationship to each specific subscale. Item distributions were analyzed including the percentage that endorsed the highest and lowest response choice for each item. Corrected item–total correlations were calculated to measure the strength of the relationship between each item and the total score of the scale. Three different levels were suggested as acceptable: 0.2, 0.3 and 0.4 [18]. We used the more liberal level of >0.2 [19] in this study so as to not exclude items of importance for some participants while less important for others due to the heterogeneity within perceived housing and among the general population of older adults.

To acquire information about the relationship among all items in a subscale, we initially calculated Cronbach’s alpha including all items for each of the subscales in the MOH and HCQ to assess internal consistency. As each domain of perceived housing consists of heterogeneous constructs, we used α > 0.5 as acceptable based on previous research in other populations [7]. A second Cronbach’s alpha we calculated for a subset of items in each subscale after removing any item that had an item–total correlation less than 0.2. Each subscale was further evaluated using descriptive statistics: mean (SD), skewness, range of scores, and floor and ceiling effects. To determine if the scales could capture variability within the population, we tested floor and ceiling effects based on the percentage of participants receiving the minimum and maximum scores on each subscale. Previous studies recommended that not more than 15–20% of participants should score at the floor or ceiling [18,20].

#### 2.3.2. Construct Validity

We utilized convergent and discriminant validity measures to determine if each scale was measuring a different aspect of perceived housing [21]. The associations among all subscales from MOH and HCQ were analyzed with Pearson correlation coefficient [22]. As both MOH and HCQ are part of the larger four-domain model of perceived housing [7], all subscales should be somewhat related. We anticipated that within each instrument, subscales would have moderate correlations and between the subscales of the two constructs, we would expect lower correlations.

## 3. Results

### 3.1. Reliability

#### 3.1.1. Meaning of Home

Table 2 shows the item analyses for the four MOH subscales. All items had at least two missing values, with the maximum of 13 missing for item 17, “Thinking about the past”. For nearly all items more than 50% of responses were for the highest response choice of 10, “strongly agree”. The exceptions were questions 7 “Having a nice day”, 16 “Not having to accommodate anyone’s wishes but my own”, 17 “Thinking about the past”, and 22 “Thinking about what living here will be like in the future”, which had 39.5%, 46.3%, 29.6%, and 16.6% of responses at the highest, respectively. Two items had >85% of participants endorsing the highest choice: 6 “Having to live in poor housing conditions” (86.1%) and 19 “Being excluded from social and community life” (86.2%). The means of items ranged from 4.9 to 9.6, and all except four items, 7, 16, 17, and 22, had a mean >8.2.

Cronbach’s alphas for all subscales of MOH were >0.50: Physical α = 0.53, Behavioral α = 0.58, Cognitive/Emotional α = 0.61, and Social α = 0.62. As seen in Table 2, corrected item–total correlations ranged from 0.08 to 0.53. Items 6 and 25 on the Physical subscale and items 8 and 27 on the Cognitive/Emotional subscale had corrected item–total correlations <0.20. These four items were removed and alpha was calculated again: Physical α = 0.61 and Cognitive/Emotional α = 0.68. The mean MOH subscale scores were all toward the high end: Physical M = 8.5 (SD = 1.20; min = 4.6, max = 10), Behavioral M = 8.7 (SD = 1.27; min = 3.0, max = 10), Cognitive/Emotional M = 8.4 (SD = 1.09; min = 3.2, max = 10), and Social M = 9.0 (SD = 1.32). All subscales except for Physical were negatively skewed (>1). However, visual inspection of the histogram for the Physical subscale indicated that most participants were at the higher end of the distribution. There were no floor effects, but some ceiling effects were present, respectively: Physical 0% and 12%, Behavioral 0% and 21.1%, Cognitive/Emotional 0% and 3.7%, and Social 0.3% and 34%.

#### 3.1.2. Housing-Related Control Beliefs

Table 3 shows the item analyses for the four HCQ subscales. All items had at least one missing value with a maximum of 22 missing on item 18, “It’s a case of luck or chance whether I will be able to continue my present way of life in my home in the future”. Participants tended to respond toward the high end (i.e., 5) on items in the Internal Control subscale, and on three items, >80% of participants chose 5, “Strongly agree”. On the External Control–Others subscale, participants tended to respond to the lowest response choice 1, “Strongly disagree”. Items 11, 17, and 23 each had >85% of participants choosing 1. Ratings on the External Control–Chance subscale were more evenly distributed, and all items had <70% of participants at ether extreme.

Cronbach’s alpha for both External Control subscales was >0.50 (Powerful Others α = 0.54, Chance α = 0.56) and was lower on the Internal Control subscale α = 0.39. As seen in Table 3, corrected item–total correlations ranged from −0.02 to 0.41. Five of eight items on the Internal Control subscale and one item on each of the External Control subscales had item–total correlations <0.2. These items were removed and alpha was calculated again: Internal Control α = 0.55, External–Powerful Others α = 0.56 and External–Chance α = 0.56. The mean HCQ subscale scores were Internal Control M = 32.8 (SD = 4.22; min = 20, max = 40), External–Powerful Others M = 14.6 (SD = 4.36; min = 8, max = 32), External–Chance M = 22.3 (SD = 5.87; min = 8, max = 37). All three subscales had relatively normal distributions, skewness <1, and there were no floor or ceiling effects, respectively: Internal Control 0% and 4.6%, External–Powerful Other 6.6% and 0%, and External–Chance 0.9% and 0%.

### 3.2. Construct Validity

As seen in Table 4, the MOH subscales had moderate positive correlations with each other, r = 0.40 to r = 0.55. Among the HCQ subscales, Powerful Others and Chance were moderately correlated, r = 0.41, while they had low to non-significant correlations with Internal Control, r = −0.05 and r = 0.14, respectively. The MOH subscales all had low to non-significant correlations with the HCQ subscales ranging from r = −0.07 to r = −0.33.

## 4. Discussion

This study aimed to assess the psychometric properties of MOH and HCQ among older adults aged 67–70 years with the overall purpose of determining if these instruments are able to measure the intended constructs in a consistent manner. All subscales of MOH and two of three subscales of HCQ had adequate psychometric properties. Consistent with previous finding in other populations [7], the Internal Control subscale indicated poor reliability, and it did not meet a satisfactory standard of validity. Furthermore, some individual items had a weak relationship with the subscales, and using a smaller set of items improved the overall reliability of these subscales.

Compared to two earlier cross-national studies that used MOH among very old adults [7,23] we identified several differences related to internal consistency. In our younger sample, internal consistency on the Physical and Behavioral subscales were somewhat lower compared to the older samples in these earlier studies, Physical (α = 0.60 and 0.69) and Behavioral (α = 0.67 and 0.67). Our further item analyses may give some clues to account for some of the differences. The MOH Physical subscale focuses on perceptions of the actual environment in the home, and two items had very weak relationships to this subscale for our sample: “Having to live in poor housing conditions” and “Being confined to rooms inside the house”. In Sweden, housing quality is quite high in comparison to many other European countries [24], and more than 85% of our sample marked the lowest possible score (strongly disagree) for this item. Hence, these items may not be as relevant for older people living in Sweden compared to the earlier study that was conducted in several European countries. Related to feeling confined in the home, our sample was younger and healthier, so it is less likely that participants would see the item as relevant for them. After removing these two items, the internal consistency was then comparable to previous studies on the Physical subscale. Differences on the Behavioral subscale were not as clear as all items had acceptable item–total correlations. Furthermore, while the Social subscale had been excluded from a previous study due to low internal consistency (α = 0.44) [7], we found an acceptable level. However, this subscale had a substantial ceiling effect with a large number of participants receiving the highest possible score, which could limit the usefulness of the subscale to discriminate among participants [18]. A similar problem also appeared for the Behavioral subscale. By increasing the heterogeneity in the sample, we would likely observe more variability among scores. Additionally, future studies with mixed age samples could be useful and allow for direct comparisons between different age groups.

Several previous studies combined the two External Control subscales of HCQ in an effort to improve internal consistency [2,7,15,23] even though the scale’s original design included them as separate, yet related, aspects of control [12]. It was also known that alpha will increase with the addition of more items even when the additional items are not highly correlated with the original set [25]. Therefore, we chose to assess the two aspects of External Control as separate constructs based on the original development of the scale. Cronbach’s alphas for both External Control subscales were acceptable, and after removing two items with low item–total correlations, no meaningful improvements were seen in internal consistency. Compared with the original German psychometric evaluation [12], we found lower levels of internal consistency for External–Powerful Others and External–Chance comparted to the two German samples, External–Powerful Others α = 0.66 to 0.72 and External–Chance α = 0.76 to 0.83. This difference was seen despite one of their samples (aged 66–69 years) being similar to our sample. This may indicate that there are cultural or contextual differences between Sweden and German that were not fully addressed when the scale was adapted for use in Sweden.

Similar to our study, other studies that have reported on the Internal Control subscale also found insufficient internal consistency [2,7,23] with the exception of the original German development paper [12]. It is worth noting that each of the other studies was conducted with populations outside of Germany or was multinational including Germany, which provides further support for the idea that there are important cultural or contextual factors related to control beliefs that differ across countries. In our Swedish sample, five of eight items on the Internal Control subscale had low item–total correlations, indicating that most of the items are not measuring the same construct. Further cultural and contextual adaptations will be necessary to make use of this subscale in future studies. Alternatively, we may consider if this subscale is necessary as low scores on the External Control subscales could be interpreted as higher internal control.

With the exception of HCQ Internal, construct validity was confirmed. As the Internal Control subscale had poor reliability, it is not surprising the validity was also poor because consistency of measurement is needed for an instrument to be valid [21]. The MOH subscales were all moderately associated with each other, which indicated that there are related aspects of the meaning of home construct but still independent aspects of that construct as the correlations were not high. The results were similar for the two HCQ External Control subscales. We had anticipated that the two External Control subscales might be highly correlated because they were combined into a single External Control scale in previous studies to improve internal consistency [2,7,15,23]. However, our results indicated that External Control–Powerful Others and External Control–Chance are two distinct aspects of external control. Divergent validity between the constructs of meaning of home and housing-related control was also confirmed, as correlations were low to non-significant. In other words, they are measuring different aspects of perceived housing within the construct of control beliefs.

The present study has several limitations. The data were collected as part of a larger study that was not designed as a psychometric study. Therefore, we were not able to fully evaluate other aspects of validity or assess other types of reliability (e.g., test retest) as it was a cross-sectional study. This also restricted our ability to test different versions (i.e., parallel forms reliability) of the questionnaires or to make direct comparisons with different age groups with our restricted range of 67–70 years-old. Other types of validity testing would also be valuable, such as face validity to see if the items seemed relevant to participants and content validity to ensure the scales covered all important aspects of the constructs. We chose relatively low thresholds for our reliability analyses, α > 0.5 and item–total correlations >0.2. While these thresholds have been noted as acceptable by some, it is more typically recommended to have α > 0.6 and preferably >0.7 as well as having item–total correlations >0.3 [18,26]. Perceived housing is a heterogeneous construct [7], and when also assessed in a heterogeneous population both factors can lead to lower values of internal consistency measures [26,27]. We therefore chose these thresholds to account for the heterogeneity. Despite these limitations, we had a large sample that allowed for a relatively robust evaluation, but further development of MOH and HCQ is recommended.

As the number of items per subscale ranged from 5–10, adding additional relevant items could improve the internal consistency. To further improve the scales, future studies should include a qualitative analysis of potential items with participants to assess the content to better address cultural and contextual issues. In addition to differences in housing quality, other important factors that are related to disparities in housing could also be considered, such as income disparities [28] and alternative housing options [29], which can be related specifically to control beliefs. Further cross-national studies are needed to evaluate how these scales discriminate between different national contexts. However, when used in a single national context such as Sweden, it could be important to evaluate and modify the content for any new national context striving to achieve conceptual equivalence [19]. Furthermore, a broad range of ages, a diversity of functional abilities, and multiple countries would allow a more robust analysis of differential validity to determine if scales are measuring different things among different groups in the heterogeneous aging population [21]. While previous work with these scales was conducted to ensure item equivalence between the original German version and translated versions used in several countries, including Sweden [7], both instruments could be further strengthened through additional development striving for conceptual equivalence [19].

## 5. Conclusions

Although we demonstrated that the MOH and HCQ scales have adequate psychometric properties to be used in Sweden, further development is recommended. While both instruments had some limitations, all subscales with one exception had adequate psychometric properties. When used in a different national context further development may be necessary to achieve conceptual equivalence. Some of the subscales could be adapted to improve reliability in the Swedish context. In the meantime, these instruments have sufficient quality to be used in exploratory studies examining the relationships between perceived housing, health and well-being.

## Figures and Tables

**Table 1 ijerph-18-04273-t001:** Characteristics of the participants (N = 371).

Variables	Number (%)
**Gender**	
Women	212 (57.1)
Men	159 (42.9)
**Age**, Mean (SD)	67.9 (0.98)
**Marital status**	
Married/cohabitating	238 (64.2)
Single	28 (7.5)
Widowed	25 (6.7)
Divorced	67 (18.1)
Committed relationship but living separately	12 (3.2)
No answer	1 (0.3)
**Education** ^1^	
Elementary school or less	139 (37.5)
Secondary school	124 (33.4)
More than secondary school	104 (28.0)
**Type of dwelling**	
Multi-dwelling	220 (59.3)
One-family house	151 (40.7)
**Years living in current dwelling:** Mean (SD)	19.2 (14.2)
**Perceived health**	
Bad	6 (1.6)
Fair	52 (14.0)
Good	115 (31.0)
Very good	124 (33.4)
Excellent	74 (19.9)
**Activities of Daily Living**	
Independent	334 (90.0)
Dependent in 1 or more activities	37 (10.0)

**^1^** Four people are missing data for education.

**Table 2 ijerph-18-04273-t002:** Analysis of the items in the four subscales of the Meaning of Home (MOH) questionnaire.

MOH Items Grouped by Subscale (Numbers Are the Order of Presentation in the Questionnaire)	% (*n*) Missing	Item Mean (SD) ^1^	% of Participants Endorsing Lowest/Highest Response (0 or 10)	Corrected Item–Total Correlations
**Meaning of home: physical aspects** **(“Being at home means for me…?”) (7 items)**				
1.Living in a place which is well-designed and geared to my needs	0.5% (2)	8.63 (1.86)	0.3%/50.9%	0.39
6. Having to live in poor housing conditions [item value was reversed for calculations]	0.8% (3)	9.62 (1.33)	1.1%/86.1%	0.17
7. Having a nice view	1.1% (4)	7.60 (2.67)	3.3%/39.5%	0.28
12. Living in a place that is comfortable and tastefully furnished	1.1% (4)	8.65 (1.88)	0.3%/52.0%	0.43
15. Feeling that home has become a burden [item value was reversed for calculations]	0.8% (3)	9.21 (1.87)	0.5%/75.5%	0.26
20. Having a base from which I can pursue activities	1.6% (6)	8.85 (2.22)	2.2%/64.9%	0.34
25. Being confined to the rooms (and things) inside the home [item value was reversed for calculations]	1.9% (7)	7.16 (3.77)	12.4%/53.6%	0.17
**Meaning of home: behavioral aspects (“Being at home means for me…?”) (6 items)**				
2. Managing things without the help of others	0.8% (3)	9.10 (1.95)	1.1%/72.8%	0.34
8. Doing everyday tasks (e.g., housework)	1.3% (5)	8.53 (2.25)	1.4%/56.0%	0.36
13. Being able to change or rearrange things as I please	1.1% (4)	8.94 (1.78)	0.3%/60.8%	0.44
16. Not having to accommodate anyone´s wishes but my own	1.6% (6)	7.56 (3.02)	4.9%/46.3%	0.25
21. No longer being able to keep up with the demands of my home (e.g., maintenance) [item value was reversed]	1.1% (4)	8.88 (2.40)	1.9%/73.6%	0.25
26. Being able to do what I please	0.8% (3)	9.11 (1.75)	0.5%/69.0%	0.38
**Meaning of home: cognitive/emotional aspects (“Being at home means for me…?”) (10 items)**				
3.Being familiar with my immediate surroundings	0.5% (2)	8.93 (1.86)	0.3%/63.7%	0.43
4. Feeling safe	0.8% (3)	9.29 (1.56)	0.3%/72.8%	0.53
9. Being bored [item value was reversed for calculations]	0.8% (3)	8.66 (2.33)	1.6%/61.1%	0.23
10. Knowing my home like the back of my hand	1.1% (4)	9.29 (1.71)	0.5%/75.7%	0.37
14. Being able to relax	0.8% (3)	9.48 (1.28)	0.3%/77.2%	0.47
17. Thinking about the past (e.g., important persons and events)	3.5% (13)	6.22 (3.57)	12.8%/29.6%	0.29
18. Enjoying my privacy and being undisturbed	0.8% (3)	9.02 (1.73)	0.5%/63.6%	0.41
22. Thinking about what living here will be like in the future	2.4% (9)	4.95 (3.69)	22.9%/16.6%	0.08
23. Feeling comfortable and cozy	0.8% (3)	9.36 (1.51)	0.5%/73.1%	0.43
27. Feeling lonely [item value was reversed for calculations]	1.3% (5)	8.56 (2.58)	2.7%/63.4%	0.17
**Meaning of home: social aspects (“Being at home means for me…?”) (5 items)**				
5. Meeting family, friends, and acquaintances	1.1% (4)	8.76 (2.30)	2.2%/65.7%	0.41
11. Living in a place where I can get no support or help from others	1.9% (7)	8.63 (2.45)	2.5%/61.8%	0.31
19. Being excluded from social and community life	0.8% (3)	9.56 (1.48)	1.4%/86.1%	0.45
24. Being able to receive visitors	0.8% (3)	9.54 (1.30)	0.5%/79.9%	0.41
28. Having a good relationship with the neighbors	1.1% (4)	8.26 (2.73)	3.8%/55.6%	0.44

^1^ Each item was rated from 0 = “strongly disagree” to 10 = “strongly agree”.

**Table 3 ijerph-18-04273-t003:** Analysis of the items in the three subscales of the Housing-Related Control Beliefs (HCQ) Questionnaire.

Housing-Related Control Beliefs (HCQ) Items Grouped by Subscale (Numbers Are the Order of Presentation in the Questionnaire)	% (n) Missing	Item Means (SD)	% of Participants Endorsing Lowest/Highest Response (1 or 5)	Items Total Correlations
**Internal control (8 items)**				
1. I have been able to set up my home in accordance with my own personal tastes and ideas	0.3% (1)	4.51 (0.88)	3.0%/65.9%	0.27
4. It depends on me whether I make use of services and facilities provided in my local areas which could make life easier	1.1% (4)	4.85 (0.51)	0.5%/89.4%	0.08
7. Everything in my home will stay the way it is, no one is going to tell me what to do	1.1% (4)	3.50 (1.46)	14.7%/34.9%	0.19
10. It is up to me to keep myself informed about new developments regarding age-friendly homes and home modification	1.9% (7)	3.88 (1.53)	17.3%/54.4%	−0.02
13. It’s up to me to take advantage of the cultural services of attractive areas in my community	0.5% (2)	4.85 (0.48)	0.3%/88.9%	0.17
16. I would never exchange the area where I live for and another living environment	0.8% (3)	3.28 (1.62)	23.9%/35.6%	0.31
19. I myself decide whose help to accept within or outside my home	0.8% (3)	4.67 (0.84)	2.4%/81.0%	0.02
22. I would not be prepared to lose the social contacts I have here in my local area by moving	2.4% (9)	3.21 (1.68)	27.6%/37.0%	0.34
**External control–Powerful others (8 items)**				
2. I rely to a great extent upon the advice of others when it comes to helpful improvements to my home	0.5% (2)	2.12 (1.21)	45.3%/4.3%	0.32
5. Whether or not I will be able to stay in my home will probably depend on other people	3.5% (13)	2.89 (1.42)	26.8%/15.6%	0.20
8. In order to do anything interesting outside of my home I have to rely on others	0.5% (2)	1.45 (1.01)	78.6%/3.0%	0.39
11. I must rely on others when it comes to making use of support services and facilities in my local area	0.3% (1)	1.29 (0.87)	87.6%/2.4%	0.41
14. When other people offer to help around the house or help me outside the home, I can’t say no	4.0% (15)	1.58 (1.08)	71.3%/4.2%	0.31
17. Other people have told me how to arrange the furnishings in my home	0.3% (1)	1.34 (0.92)	85.1%/3.0%	0.26
20. I listen to the advice of others when they tell me not to change anything in my own home	4.3% (16)	2.55 (1.45)	37.2%/12.7%	0.21
23. Other people are to blame if my home is not a place where I can enjoy life	1.1% (4)	1.24 (0.81)	89.4%/3.0%	0.04
**External control–Chance (8 items)**				
3. Having a nice place is all luck. You cannot influence it; you just have to accept it	2.2% (8)	1.63 (1.04)	65.6%/2.2%	0.26
6. It’s purely a matter of luck whether or not neighbors will step in if I need help	2.2% (8)	2.12 (1.38)	49.0%/9.1%	0.28
9. Whether or not I can stay in my home depends on luck and circumstance	2.7% (10)	4.27 (1.33)	11.4%/69.0%	0.24
12. You just have to live with the way your home is; you cannot do anything about it	0.8% (3)	2.17 (1.52)	53.0%/14.4%	0.35
15. Where and how I live has happened more by chance than anything else	0.8% (3)	2.70 (1.86)	51.1%/34.0%	0.18
18. It’s a case of luck or chance whether I will be able to continue my present way of life in my home in the future	5.9% (22)	3.28 (1.56)	24.4%/29.5%	0.36
21. The way my home has been set up just happened by chance, over time	3.2% (12)	3.90 (1.47)	15.0%/52.4%	0.30
24. Whether or not there are support services or community facilities in my neighborhood is just a matter of luck	4.6% (17)	2.29 (1.67)	55.9%/21.2%	0.21

**Table 4 ijerph-18-04273-t004:** Correlations between Meaning of Home and Housing-Related Control Beliefs subscales.

Domain of Perceived Housing	MOH				HCQ		
	Physical	Behavioral	Cognitive/ Emotional	Social	Internal	External (Powerful-Other)	External (Chance)
MOH Physical	1	0.51 **	0.51 **	0.50 **	0.20 **	0.26 **	−0.24 **
MOH Behavioral		1	0.51 **	0.40 **	0.16 **	−0.33 **	−0.06
MOH Cognitive/emotional			1	0.55 **	0.31 **	−0.07	0.08
MOH Social				1	0.27 **	−0.21 **	−0.16 **
HCQ Internal					1	−0.05	0.14 *
HCQ External (other)						1	0.41 **
HCQ External (chance)							1

* Pearson correlation is significant at the 0.05 level (2-tailed). ** Pearson correlation is significant at the 0.01 level (2-tailed). MOH = Meaning of Home Questionnaire. HCQ = Housing-Related Control Beliefs Questionnaire.

## Data Availability

Data cannot be openly shared in compliance with the original ethical approval.

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
