# Peer review of "The Psychometric Properties of the Meaning of Home and Housing-Related Control Beliefs Scales among 67–70 Year-Olds in Sweden"

_ijerph, 2021, doi:10.3390/ijerph18084273_

Round 1

Reviewer 1 Report

I have some minor concerns:

  1. Please add two more key words.
  2. Please use more scientific word, such as from line 88 to line 95, the authors used "hence" twice. And some other section, statement is pretty informal. and too much "hence"
  3. Please add knowledge gap, paper structure in the end of introduction section.
  4. Please try to add theory foundation in introduction section and cite: https://doi.org/10.3390/healthcare9010072

Author Response

Please see the attachment in the box.

Reviewer 2 Report

Thank you for making the suggested changes.

Author Response

  1. Thank you for making the suggested changes.

Response. We are happy that we were able to address all of your previous comments.

This manuscript is a resubmission of an earlier submission. The following is a list of the peer review reports and author responses from that submission.

Round 1

Reviewer 1 Report

The structure of paper is not bad. The topic is interesting. However, the biggest shortcoming is the discussion section.

The authors had results presented well. While in discussion section, they are still talking about the results. Discussion should be discussion. We need more in depth discussion on your topic not only to explore the maths results.

And why did you choose the cohort of 67-70 years old ? Without comparisons with other cohort, it's without too much significance.

Author Response

Dear reviewer 1,

We thank you for your careful consideration of our manuscript and your constructive comments, which have led to significant improvement in the manuscript. We have provided a point by point response to each reviewer comment below and made corresponding edits in the manuscript using track changes.

We look forward to having your opinion on this revision and thank you for considering the manuscript for publication.

On behalf of myself and my co-authors,

Response to the reviewer 1 comments

Comment 1: The structure of the paper is not bad. The topic is interesting. However, the biggest shortcoming is the discussion section. The authors had results presented well. While in the discussion section, they are still talking about the results. The discussion should be a discussion. We need a more in-depth discussion on your topic not only to explore the maths results.

Response 1: As the main aim of our study is the psychometric evaluation of these instruments, we have focused the discussion on comparing our results with results from previous studies among different samples/populations rather than focusing on a broader discussion around perceived aspects of housing. We appreciate your desire for a more nuanced discussion, and we have made many changes throughout the discussion including a reduction in repeating some of the results. As this was a very broad comment, these changes occur in many places through the discussion making it impractical to list all of them here.

Comment 2: And why did you choose the cohort of 67-70 years old? Without comparisons with other cohorts, it's without too much significance.

Response 2: Data were used from a larger longitudinal study aimed at examining how different aspects of the home are related to health and how this might change over time. Hence this younger cohort is being followed longitudinally. We have edited the text under Study context and participants as follows to further clarify, Lines 107-110 “…data on perceived aspects of housing was utilized from the Home and Health in the Third Age Project, which aims to study the relationships between different aspects of home and health as people age.” On line 112, we also added this text, “a longitudinal study” to show that these participants are followed over time.

Reviewer 2 Report

I think the author team has done a good job here.  Housing and well-being are undoubtedly connected, and the internal/external locus of control issue is one that has always interested me.  The author's further nuancing of external factors seems appropriate, but was not something I had considered before. 

My comments are mostly grammatical, but I would also caution the authors about a major assumption that was made from their research.  Cronbach alpha readings of .5 seems quite low to me, and I would encourage them to consider Carmines and Zeller (1979) who suggest .7 as the true threshold.  Here are my other comments:

On page 2, line 55, I am not entirely sure what “powerful others” are exactly.

On page 5, line 163, I believe the author wants to say “subset of items”.

On page 5, line 166, replace with “based on the percentage”.

On page 7, line 214, replace with “<70% of participants at either extreme”.

On page 10, line 307, replace with “data were collected”.

Author Response

Dear reviewer 2,

We thank you for your careful consideration of our manuscript and your constructive comments, which have led to significant improvement in the manuscript. We have provided a point by point response to each reviewer comment below and made corresponding edits in the manuscript using track changes.

We look forward to having your opinion on this revision and thank you for considering the manuscript for publication.

On behalf of myself and my co-authors,

Response to the reviewer 2 comments

I think the author team has done a good job here.  Housing and well-being are undoubtedly connected, and the internal/external locus of control issue is one that has always interested me.  The author's further nuancing of external factors seems appropriate but was not something I had considered before.

Comment 1: My comments are mostly grammatical, but I would also caution the authors about a major assumption that was made from their research.  Cronbach alpha readings of .5 seem quite low to me, and I would encourage them to consider Carmines and Zeller (1979) who suggest .7 as the true threshold. 

Response 1: To further clarify our reasoning we made the following change on Lines 169-171, “As each domain of perceived housing consists of heterogeneous constructs, we used α = 0.50 as acceptable based on previous research in other populations.”

We discussed this further in the limitations section, Lines 332-338 “We chose relatively low thresholds for our reliability analyses, α > 0.5 and item-total correlations > 0.2. While these thresholds have been noted as acceptable by some, it is more typically recommended to have α > 0.6 and preferably > 0.7 as well as having item-total correlations > 0.3 [18, 26]. Perceived housing is a heterogeneous construct [7], and when also assessed in a heterogeneous population both factors can lead to lower values of internal consistency measures [26, 27]. Hence, we chose these thresholds to account for the heterogeneity.

Here are my other comments:

Comment 2: On page 2, line 55, I am not entirely sure what “powerful others” are exactly.

Response 2: To clarify, we have added to the text at the first mention of powerful other, Lines 58-59 “…powerful others (i.e., family members or other people who have some level of control over aspects of housing).”

Comment 3: On page 5, line 163, I believe the author wants to say “subset of items”.

Response 3: Thank you for catching this mistake. It is changed to “subset of items” Line 172.

Comment 4: On page 5, line 166, replace with “based on the percentage”.

Response 4: This has been corrected to “based on the percentage” Line 176.

Comment 5: On page 7, line 214, replace with “<70% of participants at either extreme”.

Response 5: This has been changed to “< 70% of participants at either extreme” Line 224.

Comment 6: On page 10, line 307, replace with “data were collected”.

Response 6: This has been changed to “data were collected” Line 324.

Reviewer 3 Report

Thank you for inviting me to review this manuscript, which presents the psychometric properties of two instruments: Meaning of Home and Housing Related Control Beliefs Scales.

I found this to be a well written and engaging manuscript, which raises some very interesting points for housing research in the context of ageing. The manuscript also states the limitations of the study – namely that is not a review of all psychometric properties and that the use of these tools are dependent on cultural contexts. The authors also outlines suggested areas for further research. However, while it is well written and fitting for IJERPH, there are some areas that can be improved. If the word count is accommodating, I recommend the authors consider the following minor points:

  • 5-6 lines into the Introduction the authors suggest a lack of scales is limiting research progress in the field of perceived housing – which the authors state captures different aspects of a person’s experiences of interacting and identifying with their home environment. I think it is worth saying here, while a lack of scales might be limiting the research base, these are quite clearly concepts that better lend themselves to qualitative enquiry. I am not saying scales are not valid ways of collecting data on these areas, but the research base might be limited through other reasons – a lack of high quality qualitative research. This links in to the points that authors make about these things being open to cultural and national interpretation. The authors also suggest qualitative research be undertaken when using these scales, but I think more appreciation in the Introduction that these are qualitative concepts offers a more honest appraisal.
  • The threshold level of 0.20 when looking at reliability is on the low side (or ‘liberal’ according to the authors!). Though I understand the authors reasoning, this should be factored in as a limitation.
  • Explain in lay terms what psychometric analysis was carried out – i.e. theirs aims and purpose. Some terminology can be quite inaccessible to people who are not familiar with the niche field of psychometric analysis, but do have an interest in the field of housing and ageing. Also if a more thorough psychometric analysis was not possible, the authors should be explicit and state which areas would still benefit from further psychometric analysis
  • One of the interesting points made by the authors is how a part of the two scales may not adequately capture other national contexts, and use the example of the high quality of housing in Sweden (when compared to Germany) as impacting on the reported findings. While this point is well made by the authors, I was surprised to see little or no reference to the circumstances around older people’s housing in other European countries, both in the Introduction and Discussion. Older people’s housing and suitability is major issue in many European countries, and some reference to this in the Introduction is needed (some helpful UK related sources that I’m aware of are below). Furthermore, while the authors suggest more work is needed before these instruments should be considered for use in other national contexts, I would value more discussion about what they think the implication their findings have for the use of these instruments in other countries with different housing contexts. For example, the general poor quality and inaccessibility of the housing stock for older people in the UK is well established* – how may this impact on the configuration / use of these instruments?

*Harding, A, Hean, S, Parker, J & Hemingway, A 2020, '“It can’t really be answered in an information pack…”: A realist evaluation of a telephone housing options service for older people', Social Policy and Society, vol. 19, no. 3, pp. 361-378. https://doi.org/10.1017/S1474746419000472

*Harding, A, Parker, J, Hean, S & Hemingway, A 2018, 'Supply-side review of the UK specialist housing market and why it is failing older people', Housing, Care and Support, vol. 21, no. 2, pp. 41-50. https://doi.org/10.1108/HCS-05-2018-0006

Author Response

Dear reviewer 3,

We thank you for your careful consideration of our manuscript and your constructive comments, which have led to significant improvement in the manuscript. We have provided a point by point response to each reviewer comment below and made corresponding edits in the manuscript using track changes.

We look forward to having your opinion on this revision and thank you for considering the manuscript for publication.

On behalf of myself and my co-authors,

Response to the reviewer 3 comments

Thank you for inviting me to review this manuscript, which presents the psychometric properties of two instruments: Meaning of Home and Housing Related Control Beliefs Scales.

I found this to be a well written and engaging manuscript, which raises some very interesting points for housing research in the context of ageing. The manuscript also states the limitations of the study – namely that is not a review of all psychometric properties and that the use of these tools is dependent on cultural contexts. The authors also outline suggested areas for further research. However, while it is well written and fitting for IJERPH, there are some areas that can be improved. If the word count is accommodating, I recommend the authors consider the following minor points:

Comment 1: 5-6 lines into the Introduction the authors suggest a lack of scales is limiting research progress in the field of perceived housing – which the authors state captures different aspects of a person’s experiences of interacting and identifying with their home environment. I think it is worth saying here, while a lack of scales might be limiting the research base, these are quite clearly concepts that better lend themselves to qualitative enquiry. I am not saying scales are not valid ways of collecting data on these areas, but the research base might be limited through other reasons – a lack of high-quality qualitative research. This links into the points that authors make about these things being open to cultural and national interpretation. The authors also suggest qualitative research be undertaken when using these scales, but I think more appreciation in the Introduction that these are qualitative concepts offers a more honest appraisal.

Response 1: thank you for this thoughtful comment. We did neglect to specify that there has been some previous qualitative work in this area. We also agree that such constructs can be challenging to quantify, which relates to our ambition to evaluate and eventually improve such scales. While the qualitative studies do provide quite rich data about the relationships with the home, our future research aims to identify relationships between perceived aspects of housing and important outcomes related to health and well-being among the ageing population. We have edited the introduction to include some information related to previous qualitative work, Lines 29-33 “Some high-quality qualitative studies have been conducted capturing participants experiences via in-depth interviews and observations [4, 5, 6]. While these studies give nuanced understanding, they are usually small and do not allow generalizability on a population level. Thus, part of the challenge is the lack of adequate scales to quantify perceived aspects of housing in a larger context.”

Comment 2: The threshold level of 0.20 when looking at reliability is on the low side (or ‘liberal’ according to the authors!). Though I understand the authors reasoning, this should be factored in as a limitation.

Response 2: To further clarify our reasoning we modified this description on Lines 163-166, “We used the more liberal level of > 0.20 [16] in this study so as to not exclude items of importance for some participants while less important for others due to the heterogeneity within perceived housing and among the general population of older adults.”

We have also addressed this in the limitations section, Lines 332-338 “We chose relatively low thresholds for our reliability analyses, α > 0.5 and item-total correlations > 0.2. While these thresholds have been noted as acceptable by some, it is more typically recommended to have α > 0.6 and preferably > 0.7 as well as having item-total correlations > 0.3 [18, 26]. Perceived housing is a heterogeneous construct [7], and when also assessed in a heterogeneous population both factors can lead to lower values of internal consistency measures [26, 27]. Hence, we chose these thresholds to account for the heterogeneity.

Comment 3: Explain in lay terms what psychometric analysis was carried out – i.e. theirs aims and purpose. Some terminology can be quite inaccessible to people who are not familiar with the niche field of psychometric analysis but do have an interest in the field of housing and ageing.

Response 3: Thank you for this comment. We agree that it is important to be clearer on this point to be more accessible for the broad readership of the IJERPH. We have made several changes to address this.

We modified the objective paragraphs, Lines 99-102 “The objective of the study was to determine if two instruments about perceived aspects of housing measure the intended constructs in a consistent manner. Hence, we will evaluate the psychometric properties of the Meaning of Home Questionnaire and the Housing-Related Control Beliefs Questionnaire in a younger cohort of older adults in Sweden.”

Additional changes were made in the Data analysis section of the methods to clarify the purpose of the different analyses:

Added: Line 159 “We evaluated each item to determine the relationship to each specific subscale.”

Modified text: Line 167-168 “To get information about the relationship among all items in a subscale, we initially calculated Cronbach´s alpha including all items…”

Added: Lines 175-176 “To determine if the scales could capture variability within the population, we tested floor and ceiling effects….”

Modified text Line 180-181 “We utilized convergent and discriminant validity measures to determine if each scale was measuring a different aspect of perceived housing.”

Changed the wording in multiple places in the discussion to be less technical, for example, Lines 249-250 “This study aimed to assess the psychometric properties of MOH and HCQ among older adults aged 67-70 years with the overall purpose to determine if these instruments are able to measure the intended constructs in a consistent manner.”

Comment 4: Also if a more thorough psychometric analysis was not possible, the authors should be explicit and state which areas would still benefit from further psychometric analysis.

Response 4: In the discussion limitations paragraph we now give several examples of other types of evaluations that would be useful. Line 327-328 “test different version (i.e., parallel forms reliability) of the questionnaires”, Lines 329-332 “Other types of validity testing would also be valuable such as face validity to see if the items seem relevant to participants and content validity to ensure the scales cover all important aspects of the constructs.” We had also previously mentioned test-retest reliability on Line 326-327.

Additional recommendations are presented in the following paragraph Lines 342-360 with a specific addition, Lines 342-344 “As the number of items per subscale ranged from 5-10, adding additional relevant items could improve the internal consistency.”

Comment 5: One of the interesting points made by the authors is how a part of the two scales may not adequately capture other national contexts, and use the example of the high quality of housing in Sweden (when compared to Germany) as impacting on the reported findings. While this point is well made by the authors, I was surprised to see little or no reference to the circumstances around older people’s housing in other European countries, both in the Introduction and Discussion. Older people’s housing and suitability is a major issue in many European countries, and some reference to this in the Introduction is needed (some helpful UK related sources that I’m aware of are below).

Response 5: While housing for older adults across Europe is an important topic, the main focus of the paper was to evaluate the scales in the Swedish context with particular comparison to Germany as that is were the scales originated. We do not feel it is appropriate to address the suitability of housing in other European countries in the introduction as it was not related to the aims. We have expanded the text in the discussion to point out areas for future research in cross-national studies, Lines 346-350, “In addition to differences in housing quality, other important factors that are related to disparities in housing could also be considered such as income disparities [28] and alternative housing options [29], which can be related specifically to control beliefs. Further cross-national studies are needed to evaluate how these scales discriminate between different national contexts.”

Comment 6: Furthermore, while the authors suggest more work is needed before these instruments should be considered for use in other national contexts, I would value more discussion about what they think the implication their findings have for the use of these instruments in other countries with different housing contexts. For example, the generally poor quality and inaccessibility of the housing stock for older people in the UK is well established* – how may this impact the configuration/use of these instruments?

Response 6: Similar to the previous response we further addressed this in the discussion around future research, Lines 346-356 “In addition to differences in housing quality, other important factors that are related to disparities in housing could also be considered such as income disparities [28] and alternative housing options [29], which can be related specifically to control beliefs. Further cross-national studies are needed to evaluate how these scales discriminate between different national contexts. However, when used in a single national context, such as Sweden, it could be important to evaluate and modify the content for any new national context striving to achieve conceptual equivalence [19]. Furthermore, a broad range of ages, a diversity of functional abilities, and multiple countries would allow more robust analysis of differential validity to determine if scales are measuring different things among different groups in the heterogeneous ageing population.”
